# Deep Individual Active Learning: Safeguarding against Out-of-Distribution Challenges in Neural Networks

**DOI:** 10.3390/e26020129

**Published:** 2024-01-31

**Authors:** Shachar Shayovitz, Koby Bibas, Meir Feder

**Affiliations:** School of Electrical Engineering, Tel Aviv University, Tel Aviv 6997801, Israel; kobybibas@gmail.com (K.B.); meir@tauex.tau.ac.il (M.F.)

**Keywords:** active learning, universal prediction, deep active learning, individual sequences, normalized maximum likelihood, out-of-distribution

## Abstract

Active learning (AL) is a paradigm focused on purposefully selecting training data to enhance a model’s performance by minimizing the need for annotated samples. Typically, strategies assume that the training pool shares the same distribution as the test set, which is not always valid in privacy-sensitive applications where annotating user data is challenging. In this study, we operate within an individual setting and leverage an active learning criterion which selects data points for labeling based on minimizing the min-max regret on a small unlabeled test set sample. Our key contribution lies in the development of an efficient algorithm, addressing the challenging computational complexity associated with approximating this criterion for neural networks. Notably, our results show that, especially in the presence of out-of-distribution data, the proposed algorithm substantially reduces the required training set size by up to 15.4%, 11%, and 35.1% for CIFAR10, EMNIST, and MNIST datasets, respectively.

## 1. Introduction

In supervised learning, a training set is provided to a learner, which can then be used to choose parameters for a model that minimize the error on this set. The process of creating this training set requires annotation, where an expert labels the data points. This is a time-consuming and costly process and results in only a small subset of the data being labeled, which may not represent the true underlying model [1]. Active learning, where the training data are actively and purposely chosen, allows the learner to interact with a labeling expert by sequentially selecting samples for the expert to label based on previously observed data, thereby reducing the number of examples needed to achieve a given accuracy level [2].

Recent research has focused on obtaining a diverse set of samples for training deep learning models with reduced sampling bias. The strategies in [3,4,5,6] aim to quantify the uncertainties of samples from the unlabeled pool and utilize them to select a sample for annotation. A widely used criterion for active learning is Bayesian Active Learning by Disagreement (BALD), which was originally proposed by Houlsby et al. [3]. This method finds the unlabeled sample ${\hat{x}}_{i}$ that maximizes the mutual information between the model parameters θ and the candidate label random variable $Y_{i}$ given the candidate $x_{i}$ and training set $z^{n-1}={\left\{\left(x_{i},y_{i}\right)\right\}}_{i=1}^{n-1}$:(1)$${\hat{x}}_{i}=\underset{x_{i}}{\mathrm{argmax}}\;I\left(\theta ;Y_{i}|x_{i},z^{n-1}\right)$$
where I(X;Y|z) denotes the mutual information between the random variables X and Y conditioned on a realization *z*. The idea in BALD’s core is to minimize the uncertainty about model parameters using Shannon’s entropy. This criterion also appears as an upper bound on information-based complexity of stochastic optimization [7] and also for experimental design [8,9]. There is an issue of postulating a reasonable prior for this Bayesian approach. Empirically, this approach was investigated by Gal et al. [4], where a heuristic Bayesian method for deep learning was proposed, leading to several heuristic active learning acquisition functions that were explored within this framework.

However, BALD has a fundamental disadvantage if the test distribution differs from the training set distribution, since what is maximally informative for model estimation may not be maximally informative for test time prediction. In a previous work, Shayovitz and Feder [6] derived a criterion named Universal Active Learning (UAL) that takes into account the unlabeled test set when optimizing the training set:(2)$${\hat{x}}_{i}=\underset{x_{i}}{\mathrm{argmin}}\;I(\theta ;Y|X,x_{i},Y_{i},z^{n-1})$$
where *X* and *Y* are the test feature and label random variables. UAL is derived from a capacity–redundancy theorem [10] and implicitly optimizes an exploration–exploitation trade-off in feature selection. In addition, in the derivation of [10], the prior on θ is expressed as the capacity-maximizing distribution for $I(\theta ;Y|X,x_{i},Y_{i},z^{n-1})$. It should be noted that Smith et al. [11] have recently proposed a criterion denoted Expected Predictive Information Gain (EPIG) which also takes into account the unlabelled test set and focuses on prediction and not model estimation (In Appendix A, it is proven that EPIG is equivalent to UAL, but unlike EPIG, which does not optimize the model prior, UAL provides an expression for the optimal model prior.):(3)$${\hat{x}}_{i}=\underset{x_{i}}{\mathrm{argmax}}\;I(Y;Y_{i}|X,x_{i},z^{n-1})$$

However, the above-mentioned AL schemes assume that both training and test data follow a conditional distribution which belongs to a given parametric hypothesis class, $\left\{p\left(y|x,\theta \right)\right\}$. This assumption cannot be verified on real-world data, particularly in privacy-sensitive applications where real user data cannot be annotated [12] and the unlabeled pool may contain irrelevant information. In such cases, choosing samples from the unlabeled pool may not necessarily improve model performance on the test set. As an alternative to making distributional assumptions, we build upon the *individual setting* [13]. This setting does not assume any probabilistic connection between the training and test data. Moreover, the relationship between labels and data can even be determined by an adversary. The generalization error in this setting is known as the *regret* [14], which is defined as the log-loss difference between a learner and a *genie*: a learner that knows the specific test label but is constrained to use an explanation from a set of hypotheses. The predictive Normalized Maximum Likelihood (pNML) learner [14] was proposed as the min-max solution of the regret, where the minimum is over the learner choice and the maximum is for any possible test label value. The pNML was previously developed for linear regression [15] and was evaluated empirically for DNN [16].

The setting considered in this work, i.e., active learning with no distributional assumption, is related to the active online learning literature [17,18], which deals primarily with task-agnostic learning that does not assume a connection between the training and test tasks. The research in Yoo and Kweon [17] proposed an active learning method that works efficiently with deep networks. A small parametric module, named “loss prediction module”, is attached to a target network, and learns it to predict target losses of unlabeled inputs. Then, this module can suggest data for which the target model is likely to produce a wrong prediction. This method is task-agnostic, as networks are learned from a single loss regardless of target tasks. The research in Sinha et al. [18] suggested a pool-based semi-supervised active learning algorithm that implicitly learns a sampling mechanism in an adversarial manner. Unlike conventional active learning algorithms, this approach is task-agnostic, i.e., it does not depend on the performance of the task for which we are trying to acquire labeled data. This method learns a latent space using a variational autoencoder (VAE) and an adversarial network trained to discriminate between unlabeled and labeled data. The minimax game between the VAE and the adversarial network is played such that while the VAE tries to trick the adversarial network into predicting that all data points are from the labeled pool, the adversarial network learns how to discriminate between dissimilarities in the latent space.

Moreover, as an additional incentive for the individual setting, in scenarios involving Out-Of-Distribution (OOD) data, the application of uncertainty-based Active Learning (AL) without meticulous consideration may increase the likelihood of selecting OOD samples for labeling, surpassing the selection of in-distribution (IND) data. OOD data typically demonstrate high uncertainty, leading the AL algorithm to preferentially choose such samples for labeling, thereby inefficiently utilizing the labeling budget. Consequently, there is an urgent need for active learning methods resilient to such scenarios.

While empirical evidence has demonstrated the real-life impact of the OOD problem on AL [19], there is a scarcity of research addressing this crucial issue. The research in Kothawade et al. [20] approached OOD as a sub-task, and its sub-modular mutual information-based sampling scheme is marked by both time and memory consumption. In contrast, Du et al. [21] mandated the pre-training of additional self-supervised models like SimCLR [22], introducing hyperparameters to balance semantic and distinctive scores. The values of these hyperparameters exert a significant influence on the final performance, thereby limiting the broader applicability of the proposed approach.

In addition to the challenges highlighted in the aforementioned context, another promising avenue of research explores counterfactual training [23] to enhance OOD generalization. This approach involves learning model parameters by comparing pairs of factual samples and counterfactual samples, illustrating how changes in features lead to changes in labels. Notably, modifications to causal features and labels disrupt spurious correlations, as non-causal features are present in both factual and counterfactual samples with distinct classes [24]. Through counterfactual training, the model avoids relying on spurious correlations for predictions, enhancing its ability for OOD generalization [24,25]. This approach effectively breaks the link between non-causal features and labels, contributing to an improved OOD generalization capability. Nevertheless, counterfactual learning may be considered less feasible, as generating meaningful counterfactual samples requires sufficient and representative data, which may be challenging to obtain in some cases, especially if the dataset is limited or biased.

The research in Shayovitz and Feder [26] proposed an active learning criterion for the individual setting that takes into account a trained model, the unlabeled pool, and a small set of unlabeled test features. This criterion, denoted IAL (Individual Active Learning), is designed to select a sample to be labeled in such a way that, when added to the training set with its worst-case label, it attains the minimal pNML regret for the test set. The algorithm proposed by Shayovitz and Feder [26] for Gaussian Process Classification is based on an Expectation Propagation approximation of the model posterior. This approximation is both computationally expensive for large-scale deep neural networks (DNNs) and does not provide good enough performance in empirical tests. The computational complexity associated with the re-training for each candidate sample is extremely demanding.

### Main Contributions

Our contributions can be succinctly outlined as follows:In this investigation, we address AL in the presence of OOD challenges by utilizing a small unlabeled sample from the test distribution. We focus on the individual data setting and leverage an existing active learning criterion [26]. However, the computation of this criterion is deemed impractical for DNNs.Our primary contribution lies in the development of an efficient algorithm aimed at mitigating the challenging computational complexity associated with approximating the mentioned criterion for neural networks. Termed DIAL (Deep Individual Active Learning), this algorithm facilitates faster and more practical implementation of Individual Active Learning (IAL) for DNNs.We demonstrate that, in the presence of OOD samples, our algorithm requires only 66.2%, 91.9%, and 77.2% of labeled samples compared to recent leading methods for CIFAR10 [27], EMNIST [28], and MNIST [29] datasets, respectively, for the same accuracy level. When considering only IND samples, our approach necessitates 64.9%, 99.0%, and 64.9% labeled samples on the aforementioned datasets.In OOD scenarios, DIAL does not rely on the annotator to provide semantic information or counterfactual examples. The criterion is universally applicable across various datasets and can be implemented immediately.

This paper is organized as follows. In Section 2, the individual learning setting is introduced and the pNML is reviewed. In Section 3, IAL is presented and motivated by the minimax regret problem discussed in the previous section. In Section 4, IAL is applied to the DNN hypothesis class and a novel low-complexity algorithm denoted as DIAL is presented. In Section 5, the performance of DIAL is analyzed in comparison with state-of-the-art deep active learning algorithms. Throughout this paper, a sequence of samples will be denoted $x^{n}=\left(x_{1},x_{2},\ldots ,x_{n}\right)$. The variables x∈X and y∈Y will represent the features and labels, respectively, with X and Y being the sets containing the features’ and labels’ alphabets, respectively.

## 2. The Individual Data Setting

In the supervised learning framework, a training set consisting of *n* pairs of examples is provided to the learner:(4)$$z^{n}={\left\{\left(x_{i},y_{i}\right)\right\}}_{i=1}^{n}$$
where $x_{i}$ is the *i*-th data point and $y_{i}$ is its corresponding label. The goal of a learner is to predict an unknown test label *y* given its test data, *x*, by assigning a probability distribution $q\left(\cdot |x,z^{n}\right)$ for each training set $z^{n}$.

In the commonly used stochastic setting as defined in [13], the data follow a distribution assumed to be part of some parametric family of hypotheses. A more general framework, named *individual setting* [13], does not assume that there exists some probabilistic relation between a feature *x* and a label *y*, and so the sequence $z^{n}=\left\{x^{n},y^{n}\right\}$ is an individual sequence where the relation can even be set by an adversary. Since there is no distribution over the data, finding the optimal learner, $q\left(\cdot |x,z^{n}\right)$, is an ill-posed problem. In order to mitigate this problem, an alternative objective is proposed: find a learner $q\left(\cdot |x,z^{n}\right)$ which performs as well as a reference learner on the test set.

Denote Θ as a general index set. Let $P_{\Theta }$ be a set of conditional probability distributions:(5)$$P_{\Theta }=\left\{p\left(y|x,\theta \right)|\theta \in \Theta \right\}$$It is assumed that the reference learner knows the test label value *y* but is restricted to using a model from the given hypothesis set $P_{\Theta }$. This reference learner then chooses a model, $\hat{\theta }\left(x,y,z^{n}\right)$, that attains the minimum loss over the training set and the test sample:(6)$$\hat{\theta }=\arg\max_{\theta \in \Theta }\left[p\left(y|x,\theta \right)w\left(\theta \right)\Pi_{i=1}^{n}p\left(y_{i}|x_{i},\theta \right)\right]$$
where performance is evaluated using the log-loss function, i.e., $-\log\left(q\left(\cdot |x,z^{n}\right)\right)$.

Note that, in this work, we extended the individual setting of [30] and allowed the usage of some prior w(θ) over the parameter space, which may be useful for regularization purposes. The learning problem is defined as the log-loss difference between a learner *q* and the reference learner (genie):(7)$$R_{n}\left(q,y;x\right)=\log\frac{p\left(y|x,\hat{\theta }\right)}{q\left(y|x,z^{n}\right)}.$$

An important result for this setting is provided in Fogel and Feder [14] and provides a closed-form expression for the minimax regret along with the optimal learner, $q_{\mathrm{pNML}}$:

**Theorem** **1.**(Fogel and Feder [14]). *The universal learner, denoted as the pNML, minimizes the worst case regret:*
$$R_{n}\left(x\right)=\min_{q}\max_{y\in Y}\log\left(\frac{p\left(y|x,\hat{\theta }\right)}{q\left(y|x,z^{n}\right)}\right)$$*The pNML probability assignment and regret are:*
$$\begin{array}{cc} \; & q_{pNML}\left(y|x,z^{n}\right)=\frac{p\left(y|x,\hat{\theta }\right)}{\sum_{y}p\left(y|x,\hat{\theta }\right)}\\ & R_{n}\left(x\right)=\log\sum_{y\in Y}p\left(y|x,\hat{\theta }\right) \end{array}$$

Since the main contribution of this work relies on this theorem, we provide a short proof here:

**Proof.** We note that the regret, $R_{n}\left(x\right)$, is equal for all choices of y. Now, if we consider a different probability assignment, then it would assign a smaller probability for at least one of the possible outcomes. In this case, choosing one of those outcomes will lead to a higher regret and then the maximal regret will be higher, leading to a contradiction. □

The pNML regret is associated with the *stochastic complexity* of a hypothesis class, as discussed by Rosas et al. [31] and Zhou and Levine [16]. It is clear that for pNML, a model that fits almost every data pattern would be much more complex than a model that provides a relatively good fit to a small set of data. Thus, high pNML regret indicates that the model class may be too expressive and overfit. The pNML learner is the min-max solution for supervised batch learning in the individual setting [14]. For sequential prediction it is termed the conditional normalized maximum likelihood [32,33].

Several methods deal with obtaining the pNML learner for different hypothesis sets. The research in Bibas et al. [15] and Bibas and Feder [34] showed the pNML solution for linear regression. The research in Rosas et al. [35] proposed an NML-based decision strategy for supervised classification problems and showed that it attains heuristic PAC learning. The research in Fu and Levine [36] used the pNML for model optimization based on learning a density function by discretizing the space and fitting a distinct model for each value. For the DNN hypothesis set, Bibas et al. [37] estimated the pNML distribution with DNN by fine-tuning the last layers of the network for every test input and label combination. This approach is computationally expensive since training is needed for every test input. The research in Zhou and Levine [16] suggested a way to accelerate the pNML computation in DNN by using approximate Bayesian inference techniques to produce a tractable approximation to the pNML.

## 3. Active Learning for Individual Data

In active learning, the learner sequentially selects data instances $x_{i}$ based on some criterion and produces *n* training examples: $z^{n}$. The objective is to select a subset of the unlabelled pool and derive a probabilistic learner $q\left(y|x,z^{n}\right)$ that attains the minimal prediction error (on the test set) among all training sets of the same size. Most selection criteria are based on uncertainty quantification of data instances to quantify their informativeness. However, in the individual setting, there is no natural uncertainty measure, since there is no distribution governing the data.

As proposed in [26], the min-max regret $R_{n}$ as defined in Theorem 1 is used as an active learning criterion, which essentially quantifies the prediction performance of the training set $z^{n}$ for a given unlabeled test feature *x*. A “good” $z^{n}$ minimizes the min-max regret for any test feature and thus provides good test set performance. Since $R_{n}$ is a point-wise quantity, the average over all test data is taken:(8)$$C_{n}=\min_{x^{n}}\max_{y^{n}}\sum_{x}\log\left(\sum_{y}p\left(y|x,\hat{\theta }\right)\right)$$
where $\hat{\theta }=\hat{\theta }\left(x,y,z^{n}\right)$ is the Maximum Likelihood estimator, as defined in (Equation 6).

The idea is to find a set of training points, $x^{n}$, that minimizes the averaged log normalization factor (across unlabeled test points) for the worst possible labels $y^{n}$. This criterion looks for the worst-case scenario since there is no assumption on the data distribution. Since (Equation 8) selects a batch of points $x^{n}$, it is computationally prohibitive to solve for a general hypothesis class. In order to reduce complexity, a greedy approach denoted Individual Active Learning (IAL) is proposed in [26] which performs well empirically:(9)$$C_{n|n-1}=\min_{x_{n}}\max_{y_{n}}\sum_{x}\log\left(\sum_{y}p\left(y|x,\hat{\theta }\right)\right)$$Note that when computing (Equation 9), the previously labeled training set, $z^{n-1}$, is assumed to be available for the learner and $\hat{\theta }=\hat{\theta }\left(x,y,x_{n},y_{n},z^{n-1}\right)$. The objective in (Equation 9) is to find a single point $x_{n}$ from the unlabelled pool as opposed to the objective in (Equation 8) that tries to find an optimal batch $x^{n}$.

## 4. Deep Individual Active Learning

The DNN (deep neural network) hypothesis class poses a challenging problem for information-theoretic active learning since its parameter space is of very high dimension and the weights’ posterior distribution (assuming a Bayesian setting) is analytically intractable. Moreover, direct application of deep active learning schemes is unfeasible for real-world large-scale data, since it requires training the entire model for each possible training point. To make matters worse, for IAL, the network also needs to be trained for every test point and every possible corresponding label.

In this section, we derive an approximation of IAL for DNNs which is based on variational inference algorithms [4,38,39]. We define the hypothesis class in this case as follows:(10)$$p\left(y|x,\theta \right)=softmax\left(f_{\theta }\left(x\right)\right)$$
where θ represents all the weights and biases of the network and $f_{\theta }\left(x\right)$ is the model output before the last softmax layer. Note that *x*, *y*, and p(θ) reresent the test feature, test label, and prior on the weights, respectively.

The MAP estimation for θ is:(11)$$\hat{\theta }=\arg\max_{\theta }p\left(y^{n},y|x^{n},x,\theta \right)p\left(\theta \right),$$
where the prior p(θ) acts as a regularizer over the latent vector θ. It is common practice to use some regularization mechanism to control the training error for DNNs. In order to embed the regularization mechanism into the MAP, we introduced this prior p(θ).

Given a training set $x^{n},y^{n}$ and test couple x,y, the maximization in (Equation 11) is performed by training the DNN with all the data and converging to a steady-state maximum. Note that $x^{n-1},y^{n-1}$ are assumed to be known, while $x_{n},y_{n},x$ and *y* are not known, and all the different possibilities need to be considered, resulting in multiple training sessions of the network. In order to avoid re-training the entire network for all possible values of *x*, *y*, $x_{n}$, and $y_{n}$, we utilize the independence between soft-max scores in the MAP estimation. Using Bayes, we observe that (Equation 11) can be re-written as:(12)$$\hat{\theta }=\arg\max_{\theta }p\left(y|x,\theta \right)p\left(y_{n}|x_{n},\theta \right)p\left(\theta |y^{n-1},x^{n-1}\right)$$
where $p\left(\theta |y^{n-1},x^{n-1}\right)$ is the posterior of θ given the available data $z^{n-1}=\left(x^{n-1},y^{n-1}\right)$.

The posterior $p\left(\theta |z^{n-1}\right)$ is not dependent on the test data (x,y) and the evaluated labeling candidate $\left(x_{n},y_{n}\right)$, and thus can be computed once per selection iteration and then used throughout the IAL selection process. This is a very important point which needs to be highlighted; there is no need to re-train the network for every (x,y) and $\left(x_{n},y_{n}\right)$. We only need to train the network using $x^{n-1},y^{n-1}$ and then, during the IAL selection process, run forward passes on different θ with high $p\left(\theta |z^{n-1}\right)$ values, to compute $p\left(y|x,\theta \right)$ and $p\left(y_{n}|x_{n},\theta \right)$. This fact represents a significant computational complexity reduction since the number of possible points $x_{n}$ can be significant and we wish to avoid re-training the network for each point.

In order to acquire the weight posterior for a DNN, some advanced techniques are required [40,41,42]; these involve multiple training passes over the network. For a DNN, the posterior, $p\left(\theta |y^{n-1},x^{n-1}\right)$, is multi-modal and intractable to compute directly. Therefore, we propose approximating it by some simpler distribution, which will allow easier computation of the maximum likelihood $\hat{\theta }$.

### 4.1. Variational Inference

Variational inference is a technique used in probabilistic modeling to approximate complex probability distributions that are difficult or impossible to calculate exactly [42,43,44]. Variational inference has been used in a wide range of applications, including in Bayesian neural networks, latent Dirichlet allocation, and Gaussian processes. The goal of variational inference is to find an approximation, $q^{*}\left(\theta \right)$ from a parametric family Q, to the true distribution, $p(\theta |z^{n-1})$, that is as close as possible to the true distribution, but is also computationally tractable. This goal is formulated as minimizing the Kullback–Leibler (KL) divergence between the two distributions (also called information projection):$$q^{*}\left(\theta \right)=\underset{q\in Q}{\mathrm{argmin}}D_{KL}\left(q\left(\theta \right)||p\left(\theta |z^{n-1}\right)\right)$$

There are different algorithms for implementing variational inference; most involve optimizing a lower bound on the log-likelihood of the data under the true distribution (called evidence). The lower bound is defined as the difference between the true distribution’s data log-likelihood and the Kullback–Leibler (KL) divergence between the true distribution and the approximation. The KL divergence measures the distance between the two distributions, and so optimizing the lower bound is equivalent to minimizing the distance between the true distribution and the approximation.

One common algorithm for implementing variational inference is called mean field variational inference [45]. In this approach, the approximation to the true distribution is factorized into simpler distributions that are easier to work with, such as Gaussians or Bernoullis. The parameters of these simpler distributions are then optimized to minimize the KL divergence between the true distribution and the approximation. Another algorithm for variational inference is called stochastic variational inference [46]. In this approach, the optimization is performed using stochastic gradient descent, with a random subset of the data used in each iteration. This allows the algorithm to scale to large datasets and complex models.

### 4.2. Deep Individual Active Learning (DIAL)

In this work, we opted to use the method in Gal and Ghahramani [41], denoted as MC dropout (Monte Carlo dropout), due to its computational simplicity and favorable performance. MC dropout represents a sophisticated extension of the conventional dropout regularization technique within the domain of machine learning, and it is particularly associated with improving the robustness and uncertainty quantification of neural networks. This concept finds its roots in the broader effort to address the challenge of overfitting, a common concern in training deep learning models where the network becomes excessively attuned to the training data, hindering its generalization to new, unseen data.

Traditional dropout involves randomly deactivating, or “dropping out”, a fraction of the neurons during the training phase. This stochastic process introduces a level of noise, preventing the neural network from relying too heavily on specific features, thus enhancing its ability to generalize to diverse datasets. However, dropout is typically applied solely during the training phase, and the model’s predictions during the inference phase are based on a single deterministic forward pass through the network.

Monte Carlo dropout introduces a novel approach to the inference phase by extending the dropout mechanism beyond training. In this context, during inference, the model performs multiple forward passes with different dropout masks applied each time. This process generates a set of predictions, and the final output is obtained by averaging or aggregating these predictions. The rationale behind this technique lies in its ability to capture and quantify uncertainty associated with the model’s predictions.

By leveraging Monte Carlo dropout during inference, practitioners can gain valuable insights into the uncertainty inherent in the model’s predictions. This uncertainty is crucial in real-world applications where understanding the model’s confidence level is essential. For instance, in autonomous vehicles, medical diagnostics, or financial predictions, knowing the uncertainty associated with a model’s output can inform decision making and improve overall system reliability.

In Gal and Ghahramani [41], the authors argued that performing dropout during training on DNNs, with dropout applied before every weight layer, is mathematically equivalent to minimizing the KL divergence between the weight posterior of the full network and a parametric distribution which is controlled by a set of Bernoulli random variables defined by the dropout probability. Therefore, $p\left(\theta |y^{n-1},x^{n-1}\right)$ can be approximated in KL-sense by a distribution which is controlled by the dropout parameter. We can use this idea in order to approximate (Equation 12) and find an approximated weight distribution, $q\left(\theta \right)$. Therefore, we can re-write (Equation 12) using the variational approximation q(θ):(13)$$\hat{\theta }\approx \arg\max_{\theta }p\left(y|x,\theta \right)p\left(y_{n}|x_{n},\theta \right)q\left(\theta \right)$$

However, $q\left(\theta \right)$ as described in Gal and Ghahramani [41] is still complex to analytically compute. In fact, in Gal and Ghahramani [41], the authors do not explicitly sample from this distribution but compute integral quantities on this distribution (such as expectation and variance) using averaging of multiple independent realizations and the Law of Large Numbers (LLN). Since we focus on point-wise samples from $q\left(\theta \right)$, we cannot use the same approach as in Gal and Ghahramani [41].

In this work, we propose to sample *M* weights from $q\left(\theta \right)$ and find $\hat{\theta }$ among all the different samples. Since the weights are embedded in a high-dimensional space, the probability of the sampled weights can be assumed to be relatively uniform. Therefore, we propose approximating (Equation 13) as:(14)$$\hat{\theta }\approx \arg\max_{{\left\{\theta_{m}\right\}}_{m=1}^{M}}p\left(y|x,\theta_{m}\right)p\left(y_{n}|x_{n},\theta_{m}\right)$$

As observed by Gal and Ghahramani [41], (Equation 14) can be computed by running multiple forward passes on the network trained using dropout with $z^{n-1}$ during inference with *x* and $x_{n}$. The resulting algorithm, denoted Deep Individual Active Learning (DIAL), is shown in Algorithm 1 and follows these steps:Train a model on the labeled training set $z^{n-1}$ with dropout.For each pair of *x* and $x_{n}$, run *M* forward passes with different dropout masks and compute the product of the softmax outputs.Find the weight that maximizes DNN prediction of the test input and the unlabeled candidate input as in (Equation 12).Accumulate the pNML regret of the test point given these estimations.Find the unlabeled candidate for which the worst-case averaged regret of the test set is minimal, as in (Equation 9).

For step 2, since the variational posterior associated with MC dropout is difficult to evaluate, we assume that it is uniform for all the sampled weights.

We emphasize the significant complexity reduction provided by our approximation; a naïve implementation of pNML computation would require training the network over all possible training points $x_{n}$ and test points *x* with all possibilities of their respective labels $y_{n},y$. This would render our criterion unfeasible for real-world applications. Our proposed approach, DIAL, only requires performing training with dropout on $z^{n-1}$ only once per selection iteration and then performing forward passes (considerably faster than training passes) to obtain multiple samples of the weights.
**Algorithm 1:** DIAL: Deep Individual Active Learning**Input** Training set $z^{n-1}$, unlabeled pool and test samples ${\left\{x_{i}\right\}}_{i=1}^{N}$ and ${\left\{x_{k}\right\}}_{k=1}^{K}$.
**Output** Next data point for labeling ${\hat{x}}_{i}$
Run MC-Dropout using $z^{n-1}$ to get ${\left\{\theta_{m}\right\}}_{m=1}^{M}$
S=zeros(N,|Y|)
**for** 
*i←1* to *N* 
**do**   
**for** 
$y_{i}\in Y$ 
**do**      
**for** 
k←1 to *K* 
**do**         
Γ = 0         
**for**
$y_{k}\in Y$
**do**            
$\hat{\theta }={\mathrm{argmax}}_{\theta_{m}}p\left(y_{k}|x_{k},\theta_{m}\right)p\left(y_{i}|x_{i},\theta_{m}\right)$            
$\Gamma =\Gamma +p\left(y_{k}|x_{k},\hat{\theta }\right)$      
**end for**      
$S\left(i,y_{i}\right)=S\left(i,y_{i}\right)+\log\Gamma$
   **end for**
**end for**
${\hat{x}}_{i}={\mathrm{argmax}}_{x_{i}}\max_{y_{i}}S$


## 5. Experiments

In this section, we analyze the performance of DIAL and compare its performance to state-of-the-art active learning criteria. We tested the proposed DIAL strategy in two scenarios:The initial training, unlabeled pool, and test data come from the same distribution (IND scenario).There are OOD samples present in the unlabeled pool (OOD scenario).

The reason for using the individual setting and DIAL as its associated strategy in the presence of OOD samples is that it does not make any assumptions about the data generation process, making the results applicable to a wide range of scenarios, including PAC [47], stochastic [13], adversarial settings, as well as samples from unknown distributions.

We considered the following datasets for training and evaluation of the different active learning methods:**The MNIST dataset** [29] consists of 28 × 28 grayscale images of handwritten digits, with 60 K images for training and 10 K images for testing.**The EMNIST dataset** [28] is a variant of the MNIST dataset that includes a larger variety of images (upper and lower case letters, digits, and symbols). It consists of 240 K images with 47 different labels.**The CIFAR10 dataset** [27] consists of 60 K 32 × 32 color images in 10 classes. The classes include objects such as airplanes, cars, birds, and ships.**Fashion MNIST** [48] is a dataset of images of clothing and accessories, consisting of 70 K images. Each image is 28 × 28 grayscale pixels.**The SVHN dataset** [49] contains 600 K real-world images with digits and numbers in natural scene images collected from Google Street View.

We built upon Huang [50] and Smith et al. [11] open-source implementations of the following methods:

**The Random sampling** algorithm is the most basic approach in learning. It selects samples to label randomly, without considering any other criteria. This method can be useful when the data are relatively homogeneous and easy to classify, but it can be less efficient when the data are more complex or when there is a high degree of uncertainty.

**The Bayesian Active Learning by Disagreement (BALD)** method [4] utilizes an acquisition function that calculates the mutual information between the model’s predictions and the model’s parameters. This function measures how closely the predictions for a specific data point are linked to the model’s parameters, indicating that determining the true label of samples with high mutual information would also provide insight into the true model parameters.

**The Core-set** algorithm [5] aims to find a small subset from a large labeled dataset such that a model learned from this subset will perform well on the entire dataset. The associated active learning algorithm chooses a subset that minimizes this bound, which is equivalent to the k-center problem.

**The Expected Predictive Information Gain (EPIG)** method [11] was motivated by BALD’s weakness in prediction-oriented settings. This acquisition function directly targets a reduction in predictive uncertainty on inputs of interest by utilizing the unlabelled test set. It is shown in Appendix A that this approach is similar to UAL [6], where the main difference is that UAL assumes the stochastic setting, where the data follow some parametric distribution.

### 5.1. Experimental Setup

The first setting we consider consists of an initial training set, an unlabeled pool (from which the training examples are selected), and an unlabeled test set, all drawn from the **same distribution**. The experiment includes the following four steps:A model is trained on the small labeled dataset (initial training set).One of the active learning strategies is utilized to select a small number of the most informative examples from the unlabeled pool.The labels of the selected samples are queried and added to the labeled dataset.The model is retrained using the new training set.

Steps 2–4 are repeated multiple times, with the model becoming more accurate with each iteration, as it is trained on a larger labeled dataset.

In addition to the standard setting, we evaluate the performance in **the presence of OOD samples**. In this scenario, the initial training and test sets are drawn from the same distribution, but the unlabeled pool contains a mix of OOD samples. When an OOD unlabeled sample is selected for annotation, it is not used in training of the next iteration of the model. Across all x-axis values in the subsequent test accuracy figures, the presented metric is the count of Oracle calls, reflecting the instances when a selection strategy chose a sample, whether it be IND or OOD. It is crucial to differentiate this metric from the training set size, as the selection of an OOD sample leads to an increase in the number of Oracle calls, while the training set size remains unaffected. An effective strategy would recognize that OOD samples do not improve performance on the test set and avoid selecting them.

A visual representation of the scenario with OOD samples is illustrated in Figure 1a–c. These figures show the unlabeled pool, which contains a mixture of both IND and OOD samples. Figure 1d–f show the test set, which contains only IND samples. We argue that this is a representative setting for active learning in real life. In the real world, unlabelled pools are collected from many data sources and will most certainly contain OOD data. The process of pruning the unlabelled pool is a costly process and involves human inspection and labeling, which needs to be minimized. This is exactly the goal of active learning and finding a criterion which implicitly filters OOD data is of significant interest.

### 5.2. MNIST Experimental Results

Following Gal et al. [4], we considered a model consisting of two blocks of convolution, dropout, max-pooling, and ReLu, with 32 and 64 5 × 5 convolution filters. These blocks are followed by two fully connected layers that include dropout between them. The layers have 128 and 10 hidden units, respectively. The dropout probability was set to 0.5 in all three locations. In each active learning round, a single sample was selected. We executed all active learning methods six times with different random seeds. For BALD, EPIG, and DIAL, we used 100 dropout iterations and employed the criterion on 512 random samples from the unlabeled pool. MNIST results are shown in Figure 2a. The largest efficiency is at a number of Oracle calls of 71, where DIAL attains an accuracy rate of 0.9, while EPIG and BALD achieve an accuracy rate of 0.86.

To simulate the presence of OOD samples, we added the Fashion MNIST to the unlabeled pool such that the ratio of Fashion MNIST to MNIST was 1:1. In this setting, DIAL outperforms all other baselines, as shown in Figure 2b. DIAL is the top-performing method and has better accuracy than EPIG, BALD, Core-set, and Random. The largest efficiency is an accuracy rate of 0.95, where DIAL uses 240 Oracle calls, while BALD needs 307 (−35.1%). EPIG never reaches this accuracy level. The number of Oracle calls for additional accuracy rates is shown in Table 1.

### 5.3. EMNIST Experimental Results

We followed the same setting as the MNIST experiment with a slightly larger model than MNIST consisting of three blocks of convolution, dropout, max-pooling, and ReLu. The experimental results, shown in Figure 3a, indicate that DIAL is the top-performing method. For an accuracy rate of 0.56, it requires 8.3% less Oracle calls when compared to the second best method.

In the presence of OOD samples, the DIAL method outperforms all other baselines, as shown in Figure 3b and Table 2. For 300 Oracle calls, DIAL achieves a test set accuracy rate of 0.52, while BALD, EPIG, Core-set, and Random attain 0.51, 0.5, 0.42, and 0.40, respectively. For an accuracy rate of 0.53, DIAL needs 308 Oracle calls, while BALD and EPIG require 346 and 342, respectively (−11%). Moreover, Core-set and Random do not achieve this accuracy.

### 5.4. Cifar10 Experimental Results

For the CIFAR10 dataset, we utilized ResNet-18 [51] with an acquisition size of 16 samples. We used 1K initial training set size and measured the performance of the active learning strategies up to a training set size of 3K. The CIFAR10 results are shown in Figure 4a. Overall, DIAL and Random perform the same and have a better test set accuracy than the other baselines for Oracle calls greater than 2100.

When the presence of OOD samples in the unlabeled pool is considered, as shown in Figure 4b, DIAL outperforms the other methods. Table 3 shows the number of Oracle calls required for different accuracy levels. For the same accuracy rate of 0.65, DIAL needs up to 15.4% less Oracle calls than the second best method. This can be explained by Figure 5, which shows the ratio of OOD samples to the number of Oracle calls. The figure suggests that DIAL outperforms other criteria by selecting fewer OOD samples, contributing to its commendable performance. It is noteworthy that in all OOD scenarios, DIAL demonstrated superior ability to identify in-distribution samples without explicit knowledge of the distribution and solely utilizing unlabeled test features. This underscores the universality of DIAL, showcasing its adaptability to various distribution shifts. Additionally, the second-best performer, EPIG, also considers the unlabeled test set and performs better than other baseline methods but falls short of DIAL. Notably, BALD and Core-set exhibit similar behavior, possibly attributed to their emphasis on model estimation rather than leveraging the test set for predictive focus.

## 6. Limitations

The proposed DIAL algorithm is a min-max strategy for the individual settings. However, DIAL may not be the most beneficial approach in scenarios where the unlabeled pool is very similar to the test set, where different selection strategies may perform similarly and with smaller complexity. This limitation of DIAL is supported by the experimental results of Section 5.4, where the DIAL algorithm performed similarly to random selection for the CIFAR10 dataset (but better than all the other baselines).

Another limitation of DIAL is that it has a higher overhead computation compared to other active learning methods such as BALD. This is because DIAL involves computing the regret on the test set, which requires additional computations and could become significant when the unlabeled pool or the test set are very large.

## 7. Conclusions

In this study, we propose a min-max active learning criterion for the individual setting, which does not rely on any distributional assumptions. We have also developed an efficient method for computing this criterion for DNNs. Our experimental results demonstrate that the proposed strategy, referred to as DIAL, is particularly effective in the presence of OOD samples in the unlabeled pool. Specifically, our results show that DIAL requires 12%, 10.4%, and 15.4% fewer Oracle calls than the next best method to achieve a certain level of accuracy on the MNIST, EMNIST, and CIFAR10 datasets, respectively.

As future work, we plan to investigate batch acquisition criteria that take into account batch selection. This will allow us to consider the relationship between the selected samples and the overall composition of the batch, which may lead to even further improvements in performance.

## Figures and Tables

**Figure 1 entropy-26-00129-f001:**
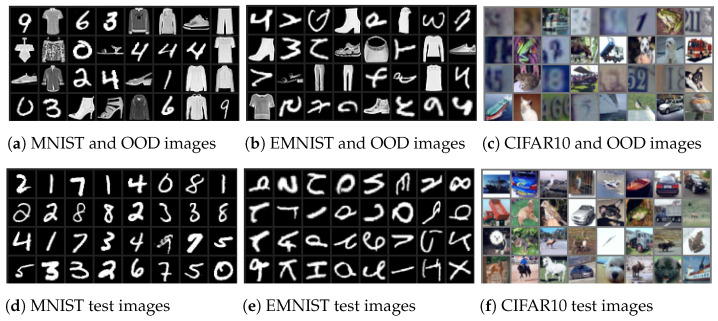
Datasets that contain a mix of images with OOD samples. (Top) Unlabeled pool contains OOD samples (Bottom). Test set includes only valid data.

**Figure 2 entropy-26-00129-f002:**
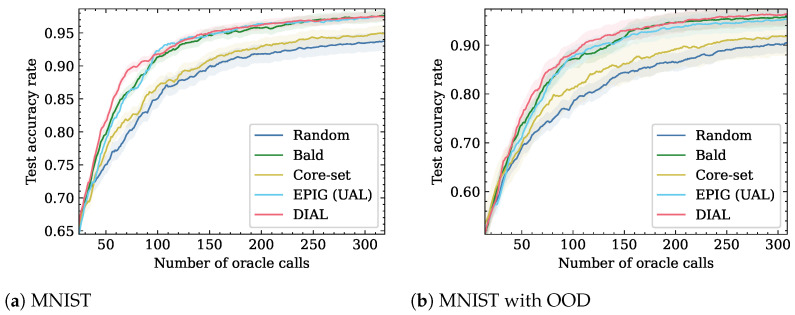
Accuracy as function of number of Oracle calls on MNIST dataset. DIAL outperforms the baselines for the two setups.

**Figure 3 entropy-26-00129-f003:**
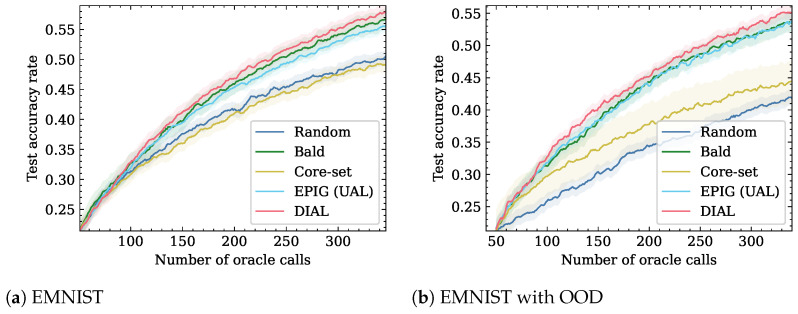
Active learning performance on the EMNIST dataset. DIAL is more efficient than tested baselines in the number of Oracle calls.

**Figure 4 entropy-26-00129-f004:**
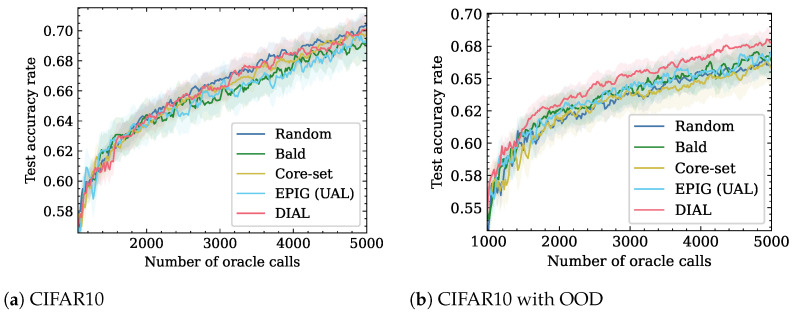
The left figure illustrates the performance of CIFAR10 using only IND samples. The DIAL method performs similarly to the Random method. The figure on the right shows the performance of a combination of OOD samples, where DIAL outperforms all other methods.

**Figure 5 entropy-26-00129-f005:**
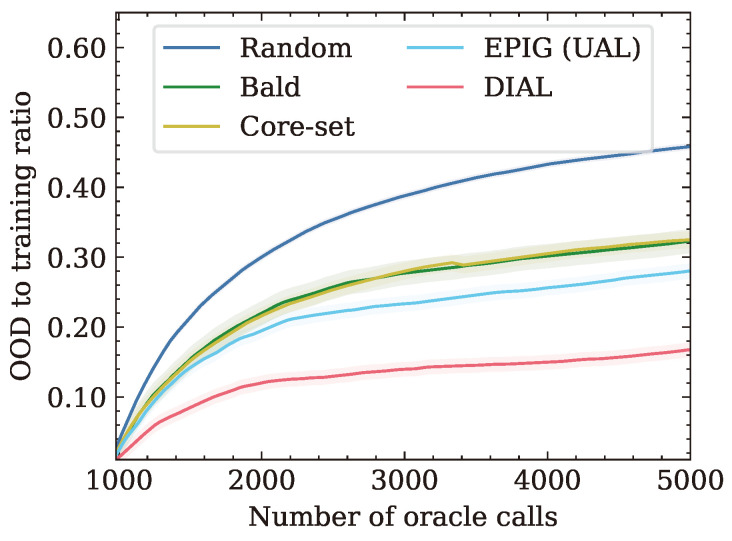
The amount of chosen OOD samples for CIFAR10 with the presence of OOD samples.

**Table 1 entropy-26-00129-t001:** MNIST with OOD number of Oracle calls at x% accuracy.

Methods	85% Acc.	75% Acc.	65% Acc.
Random	145	73	36
Core-set	117	61	33
BALD	83	51	32
EPIG	84	56	35
DIAL	**73 (−12.1%)**	**48 (−5.9%)**	**30 (−6.2%)**

**Table 2 entropy-26-00129-t002:** EMNIST with OOD number of Oracle calls at x% accuracy.

Methods	40% Acc.	30% Acc.	25% Acc.
Random	281	140	80
Core-set	221	96	62
BALD	154	85	**59**
EPIG	157	**84**	**59**
DIAL	**138 (−10.4%)**	**84 (−1.2%)**	**59 (0%)**

**Table 3 entropy-26-00129-t003:** CIFAR10: the presence of OOD samples: Number of Oracle calls at specific accuracy rate values.

Methods	66% Acc.	62% Acc.	58% Acc.
Random	3956	1828	1220
Core-set	4468	1844	1412
BALD	4020	1636	1202
EPIG	3636	1700	1108
DIAL	**3076 (−15.4%)**	**1556 (−4.9%)**	**1060 (−4.3%)**

## Data Availability

The paper itself contains all the information required to assess the conclusions. Additional data related to this paper may be requested from the corresponding author.

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
