# Peer review of "Deep Individual Active Learning: Safeguarding against Out-of-Distribution Challenges in Neural Networks"

_entropy, 2024, doi:10.3390/e26020129_

Round 1
Reviewer 1 Report
Comments and Suggestions for Authors
Dear authors,
That work proposes a method for tackling a possible mismatch between the data distributions of labeled and unlabeled data during the active learning framework. The authors have contributed to that field in the past, augmenting or adopting here previous approaches and procedures, as well as claiming the functionality and the limitations of related methods. Some theoretical proofs are also discussed here, supporting the logic that they followed in order to suggest that innovation (DIAL).
In general, the structure of that manuscript is good, explaining the big picture of active learning field, and discussing recent advancements. The main focus is given on the integration of neural networks in that part, ignoring previous milestones (e.g., methods applied to tabular dataset).
The experimental procedure simulates the necessary setup for extracting useful information regarding the proposed algorithm. My main concerns are the next ones:
- I understand that the labeled / unlabeled instances present different distribution, rather than the actual test set. If this is the case, please correct the points that mention the latter case.
- Text-based datasets should be examined, where the distribution of the generated embeddings could be quite irrelevant for non-related segments / phrases.
- The limitation of the small test set due to the computational overhead of the proposed algorithm should be discusses in further depth. In several applications, the test set is quite larger than the training set (L alone, or even L+U subsets). If the time response of your algorithm is ~1000x slower than that of the rest methods in such cases, the spectrum of the real-life applications where you can wait for user input too long is really limited. This reduces the impact of measuring performance in specific oracles' calls. You have to provide further details and discussion in that point.
- Since you compare your methods using open-source datasets with other approaches that can be easily reproduced, I was expecting to see a code repo implementing your method and generating those results.
- The discussion of the 3 different datasets and their results in different subsections could be avoided, by aggregating them into one that summarizes the main points.
- The improvement of the proposed algorithm against the 2n best into the matrices is not useful.9
- I would also discuss or depict some representative examples of the OOD images that were selected from the proposed and the rest methods so as to understand what has happened.
Comments on the Quality of English Language
Clear and concise writing style.
Author Response
All the comments to the reviewer are in the attached pdf.

Reviewer 2 Report
Comments and Suggestions for Authors
This paper proposes an efficient algorithm which can address the challenging computational complexity associated with approximating this criterion for neural networks. The study proposes a min-max active learning criterion for the individual setting, which does not rely on any distributional assumptions. This study also developed an efficient method for computing this criterion for DNNs. The content organization is reasonable and the text content is rich, but it is suggested that the author should modify the following problems to improve the quality of the paper.
1.The introduction of the first section should introduce the background of the article, the existing problems and the solutions proposed in this article. The content of the first section of this article includes related work, which is inconsistent with the title. In the last paragraph of the first section, the author should describe his innovation points in order to make the structure of the article clearer.
2.In the first section, most of the documents cited by the author are older ones. Have there been no articles published in relevant fields in recent years? Please quote some articles with high timeliness.
3.Why not start the serial numbering with the three formulas at the beginning of the third paragraph of the introductory section?
4.The proof of Theorem 1 formulae is discussed only through some linguistic theories and lacks substantive formulae proofs.
5.The layout of the article needs to be readjusted, besides, some content in the article is not smooth, and there should be a causal relationship between sentences. It is suggested to check the sentence problem after translation.
6.The description about the research is relatively complete, but it lacks relevant ablation experiments.In addition, it is suggested to supplement the comparative experiments with the more latest methods to increase the persuasiveness of the paper.
Comments on the Quality of English LanguageN/A
Author Response

(The authors gave the same response as above.)
